# Dietary Interventions with or without Omega-3 Supplementation for the Management of Rheumatoid Arthritis: A Systematic Review

**DOI:** 10.3390/nu13103506

**Published:** 2021-10-04

**Authors:** Tala Raad, Anne Griffin, Elena S. George, Louise Larkin, Alexander Fraser, Norelee Kennedy, Audrey C. Tierney

**Affiliations:** 1Discipline of Dietetics, School of Allied Health, Faculty of Education and Health Sciences and Health, University of Limerick, V94 T9PX Limerick, Ireland; anne.griffin@ul.ie (A.G.); audrey.tierney@ul.ie (A.C.T.); 2Implementation Science and Technology Cluster Health Research Institute, University of Limerick, V94 T9PX Limerick, Ireland; louise.larkin@ul.ie (L.L.); Norelee.Kennedy@ul.ie (N.K.); 3Institute for Physical Activity and Nutrition (IPAN), School of Exercise and Nutrition Sciences, Deakin University, Geelong, VIC 3220, Australia; elena.george@deakin.edu.au; 4Discipline of Physiotherapy, School of Allied Health, Faculty of Education and Health Sciences, University of Limerick, Republic of Ireland University of Limerick, V94 T9PX Limerick, Ireland; 5Department of Rheumatology, University Hospital Limerick, V94 T9PX Limerick, Ireland; alexander.fraser@hse.ie; 6Graduate Entry Medical School, Faculty of Education and Health Sciences, University of Limerick, V94 T9PX Limerick, Ireland; 7School of Allied Health, Human Services and Sport, Faculty of Science and Engineering, La Trobe University, Melbourne, VIC 3086, Australia

**Keywords:** dietary interventions, rheumatoid arthritis, omega-3 supplements, nutrition, systematic review

## Abstract

Rheumatoid Arthritis (RA) is a chronic autoimmune condition characterized by symptoms of inflammation and pain in the joints. RA is estimated to have a worldwide prevalence of 0.5–1%, with a predominance in females. Diet may play an important role in the symptoms of RA; however, little is known about the effects of various diets. The aim of this systematic review is to explore the effect of dietary interventions, with or without omega-3 supplementation for the management of RA. The electronic databases MEDLINE, EMBASE, CINAHL, and the Cochrane Library were systematically searched for clinical trials investigating dietary interventions, with or without omega-3 supplementation to retrieve papers from inception to April 2021. Randomized and non-randomized controlled trials of dietary interventions in adults with RA were eligible for inclusion. Twenty studies with a total of 1063 participants were included. The most frequently reported outcomes were pain, duration of morning stiffness, joint tenderness, grip strength and inflammatory markers. Dietary interventions with an anti-inflammatory basis may be an effective way for adults with RA seeking complementary treatments, potentially leading to improvements in certain parameters. However, there is a need for longer duration studies that are well-designed and sufficiently powered to investigate the influence of diet on RA.

## 1. Introduction

Rheumatoid arthritis (RA) is a chronic inflammatory disease that affects almost 0.5–1% of the population globally [1]. RA is the most prevalent form of inflammatory polyarthritis and is three times more common in women compared to men [2]. RA occurs when the immune system mistakes the body’s cells for external invaders and releases inflammatory substances that attack the lining of the joints. Symptoms of RA may include pain, joint stiffness, swelling, fatigue and weakness [3]. RA affects nearly all organs in the body leading to comorbid conditions [4] such as cardiovascular diseases, gastrointestinal disorders, infections, osteoporosis and depression [5]. The prevalence of comorbid conditions reported in different studies varies between 40 and 66% [6,7]. Treatment for RA involves lifelong pharmacological adherence to delay the advancement of the disease, control symptoms and maintain the person’s ability to function [8]. The most commonly prescribed medications include nonsteroidal anti-inflammatory drugs (NSAIDs), corticosteroids and disease-modifying anti-rheumatic drugs (DMARDs) to decrease joint pain, swelling, and inflammation [9].

The pathogenesis of RA remains unclear; genetic predisposition represents a great percentage of risk while the remainder is thought to be connected to modifiable factors such as tobacco smoking, diet and exercise [10]. Diet is a major modifiable determinant of chronic conditions with a large body of evidence showing that modifications to improve diet quality are directly associated with health benefits [11]. Diet is an area of interest for people living with RA as a way of improving symptoms [12]. However, the effect of various dietary interventions in RA remains controversial and inconclusive within existing literature. Despite the number of trials that have explored different diet and nutrient supplementation approaches, specific dietary recommendations and evidence-based dietary guidelines for this population are lacking. A recent systematic review concluded that the evidence for the effects of diets and dietary supplements on the Disease Activity Score in 28 joints (DAS28) in people with RA is insufficient and conclusions may not be drawn [13]. Similarly, another review evaluating the effects of diets, dietary supplements, and fasting in RA established that the effectiveness of and need for diets and dietary supplements in RA remains unclear as the responses to diets and supplements vary from one person to another [14]. Conclusions from the two recently published systematic reviews are in line with findings from a Cochrane review published in 2009 on the effectiveness and safety of dietary interventions in RA [15].

Omega-3 polyunsaturated fatty acids (PUFAs) include eicosapentaenoic acid (EPA) and docosahexaenoic acid (DHA) which are mainly derived from oily fish and fish oil supplements and alpha-linoleic acid (ALA) derived from plant sources [16]. The effect of marine omega-3 PUFAs on the functional responses of cell types involved in inflammation has been researched for many years [17]. Omega-3 PUFAs regulate signaling pathways of anti-oxidants and alter inflammatory pathways by competing with omega-6 PUFAs which are transformed to pro-inflammatory eicosanoids [18]. Omega-3 PUFAs are the most studied supplements in RA with several clinical trials conducted among adults over the years [19,20]. A recently published systematic review [21] concluded that supplementation with omega-3 PUFAs led to substantial improvements in the duration of early morning stiffness (EMS), pain levels, erythrocyte sedimentation rate (ESR), physical function, grip strength, joint tenderness and levels of leukotriene B4 (LTB4). Given the evidence relating inflammation to disease progression, omega-3 PUFAs play a significant role through modulation of the inflammatory processes and pathways.

Although multiple systematic reviews have explored the effectiveness of dietary interventions in RA [13,14,15], no systematic review has focused explicitly on dietary interventions either alone or in combination with omega-3 PUFAs supplementation in this area. Therefore, the aim of this systematic review is to examine the effects of different dietary interventions, with or without omega-3 supplementation for the management of RA.

## 2. Materials and Methods

This systematic review adheres to the relevant criteria of the Preferred Reporting Items for Systematic reviews and Meta-Analyses (PRISMA) statement (Appendix A) [22] and the Cochrane Handbook for Systematic Reviews of Interventions [23]. The review was registered in PROSPERO, the international prospective register of systematic reviews (CRD42020147415).

### 2.1. Search Terms and Strategy

A systematic search for all relevant articles was performed using the electronic databases MEDLINE, EMBASE, CINAHL, and the Cochrane Library on 12 April 2021. English language limits were applied; there were no restrictions on years. A comprehensive search string was developed using Boolean operators with variations of key search terms including Rheumatoid Arthritis, diet or nutrient interventions and omega-3 fatty acids or fish oil. This search string was finalized prior to commencing the literature search (Appendix A). The search was not restricted to specific outcomes in order to ensure all relevant literature exploring the effect of dietary interventions on different aspects of RA management were included. Additionally, a comprehensive examination of reference lists of included articles was conducted.

### 2.2. Eligibility Criteria

The Patient, Intervention, Comparators, Outcome, and Study Design (PICOS) method was used to develop the inclusion and exclusion criteria (shown in Table 1). All randomized (RCTs) and non-randomized clinical trials (NRCTs) that investigated the effect of dietary interventions with or without omega-3 supplementation in adults with RA were included. Populations included adults (>18 years) diagnosed with RA, in accordance with a specified criterion e.g., American Rheumatology Association (ARA) criteria [24] or ACR/EULAR 2010 rheumatoid arthritis classification criteria [25]. Studies had to include outcome measures related to aspects of RA management such as symptom control, clinical and biochemical measures, or disease activity. Any dietary intervention approach or delivery method was included. Where two reports shared the same patient group, the report with the most relevant outcome measures was included in this review in order to avoid duplication.

### 2.3. Study Selection Process

All references were imported into a bibliographic database in order to automatically remove duplicates (EndNote X8). Studies were manually screened by title/abstract for inclusion by the lead author (TR) to identify studies that potentially fulfilled the inclusion criteria. Full-text articles of these studies were retrieved and two reviewers (TR, AG) independently applied the inclusion/exclusion criteria to papers identified. Discrepancies were managed by discussion to reach consensus.

### 2.4. Data Extraction

Data extraction was conducted by the lead author (TR). For each article that met the inclusion criteria, the following parameters were extracted: (1) author/date, (2) country, (3) study design, (4) sample size, (5) population characteristics—percentage of females, age, (6) type of dietary intervention, (7) comparator, (8) intervention duration, (9) outcome measures, and (10) main results. Disease activity and disease duration were not documented since not all studies reported on these parameters.

### 2.5. Quality Assessment

The included studies were independently assessed by two reviewers (TR, AG) using the Cochrane Collaboration’s tool for assessing the Risk of Bias in Randomized Studies (ROBIS) [26] and the Risk Of Bias In Non-randomized Studies—of Interventions (ROBINS) [27]. Both the ROBIS and ROBINS focus on a study’s internal validity. The Cochrane risk-of-bias tools provide a comprehensive framework organised into several domains to assess the quality of studies. These domains evaluate the risk of bias arising from the randomization process, deviations from intended interventions, missing outcome data, outcome measurements, and selection of reported results. During the assessment, each domain is given a rating of low, high, or unclear. Conflicts in quality assessment were resolved through consensus.

### 2.6. Data Analysis

Due to the heterogeneity of the dietary interventions, inconsistency between the study groups and diversity of outcome measures, a meta-analysis was not conducted. This review is descriptive in nature; comparisons were made separately based on the type of dietary intervention. Both within-group and between-group analyses were considered narratively for the purpose of this review.

## 3. Results

### 3.1. Study Selection

A total of 3370 articles were retrieved from the database searches, 2921 articles remained after duplicates were removed of which 327 full-text articles were assessed for eligibility. In total, twenty studies met the inclusion criteria and were included in this systematic review. The search strategy and selection process are reported as per the PRISMA flowchart [22] and presented in Figure 1. Additionally, the reference lists of searched articles were screened to identify any potential studies; however, no further articles were retrieved. Of the 20 studies included, 18 were randomized controlled trials (RCTs) and two were non-randomized controlled trials (NRCTs) [28,29]. A summary of the included studies’ characteristics is shown in Table 2.

### 3.2. Study and Participant’s Characteristics

The included studies were published across a wide timeframe (1979–2020) and provide results from a total of 1063 participants, with more than 80% females and representing nine different countries—Sweden [30,41,42,45], UK [32,36,44], Italy [40], Norway [28,29,34], Denmark [37,39], Finland [38], Netherlands [35], US [31] and Germany [43,47]. The number of participants ranged from 12 to 130 in the RCTs. The mean age of participants was 48.5 years. One study did not report on participants’ age [32]. The dietary intervention period ranged from 3 weeks to 13 months with a mean duration of 18.2 weeks. The studies used different criteria for the diagnosis of RA including 1987 ACR criteria [48] and ARA criteria [49].

### 3.3. Intervention Characteristics

The included studies present distinct dietary interventions. Of the twenty studies included in this review, two studies intervened with a Mediterranean type dietary pattern, one study compared a Cretan Mediterranean diet to habitual diet [42] and the other compared a Mediterranean diet to a healthy diet [44]. Four studies intervened with a vegan diet [38,41,45,47]. Two studies intervened with fasting for 7–10 days followed by one year vegetarian diet for the remainder of the study period [30,34]. Three studies intervened with an elemental diet provided in the form of an easily digestible liquid formula [28,36,39] and another four studies intervened with allergen-free diets by eliminating certain foods that commonly cause allergies such as wheat, eggs, dairy products and spice [31,32,35,40] Two studies intervened with an anti-inflammatory diet rich in omega-3 PUFAs [43,46]. One study compared a ketogenic diet to 7-day fasting [29], one study compared a diet high in polyunsaturated fatty acids (PUFAs) to a diet high in saturated fatty acids [33] and one study compared an energy adjusted diet to habitual diet [37]. Only two studies included omega-3 PUFAs supplementation with the dietary intervention. One study was a double-blind crossover study whereby participants in both study groups were assigned to receive either placebo or fish oil capsules (30 mg/kg body weight) [43] and the latter included a diet high in PUFAs and complemented with omega-3 supplements providing 1.6 g EPA and 1.1 g DHA per day [33].

### 3.4. Risk of Bias within Studies

The quality assessment of studies using the Cochrane risk of bias tool is presented in Figure 2. The quality varied between the studies. The majority of studies had a high or unclear risk of bias, which decreases the quality of evidence in the included studies. The main source of bias identified across studies was selection bias as the randomized sequence generation and the concealment of allocations prior to intervention assignment was unclear. Six studies were identified as being at risk of performance bias also, due to the lack of blinding of outcomes assessors. Based on these assessments, only three studies were found to be of low risk of bias for most domains. We included all studies in this review including those deemed to be of high risk of bias as not including them could alter the results and conclusions of this review.

### 3.5. Outcome Measures

The following presents the cumulative results of the dietary interventions on reported outcome measures presented by type of outcome measure.

#### 3.5.1. Inflammatory Markers

##### Erythrocyte Sedimentation Rate

Of the twenty included studies, eight reported on ESR levels. Two studies reported significant reductions in ESR levels. Haugen et al. [28] reported that an elemental diet followed for 3 weeks significantly reduced ESR (*p* = 0.03). Similarly, fasting for 7–10 days followed by a vegetarian diet for one year lead to a decrease in ESR levels (*p* < 0.002) [34].

##### C-Reactive Protein

Of the seven studies that evaluated CRP, two studies reported significant improvements post-intervention. Adam et al. [43] reported that participants following both an anti-inflammatory diet and a Western diet, who were on methotrexate and were supplemented with fish oil experienced significant improvements in CRP levels after 3 months (2.03 ± 1.8 mg/dL vs. 1.69 ± 1.5 mg/dL) (*p* < 0.05). CRP also significantly improved with 7–10 days fasting followed by a vegetarian diet for one year (*p* < 0.005) [34].

##### Tumor Necrosis Factor-α

One study reported on the effects of a dietary intervention on levels of TNF- α. The 8-month double-blind cross-over study found a significant decrease in TNF- α following both an anti-inflammatory diet and Western diet with fish oil supplementation for months 6, 7, 8 (*p* = 0.004) [43].

##### Platelet Count

Platelet count was significantly reduced following both an elemental diet for 3 weeks (*p* = 0.02) [28] and with 7–10 days fasting was followed by a one year vegetarian diet (*p* < 0.006) [34].

##### Leukotriene 4

Adam et al. reported significant reductions in LTB4 levels following an anti-inflammatory diet with fish oil supplementation for 3 months (*p* = 0.009) [43].

##### Interleukin-6 and Interleukin-10

One Study Reported That IL-6 Decreased Significantly after a 7-Day Fast (35.5 to 22.5 pg/mL) (*p* < 0.05) [29]. No significant changes were reported for IL-10 in any of the included studies.

##### Dehydroepiandrosterone Sulfate

Fraser et al. reported that participants in both the ketogenic and the fasting diet groups experienced a 34% increase in serum DHEAS levels after 7 days compared with baseline values (*p* < 0.006) [29].

##### Oxidized Low-Density Lipoprotein (OxLDL)

One study found a significant decrease in OxLDL in participants following a gluten-free vegan diet (54.7 to 48.6) (*p* = 0.09) [45].

##### Immunoglobulins

Hafstrom et al. [41] compared the effects of a gluten- free vegan diet to a well-balanced non-vegan diet. Investigators found that IgG anti-gliadin decreased significantly in the vegan diet group (5 to 2) (*p* = 0.0183) while IgA anti-gliadin decreased significantly in the non-vegan diet group (14.5 to 12.5) (*p* = 0.0201). Moreover, Lederer et al. [47] found that sialylated antibodies increased significantly in both the vegan diet group (0.8 ± 0.4 to 1.4 ± 1.4) (*p* = 0.023) and in the meat rich group (0.9 ± 0.5 to 1.6 ± 1.2) (*p* = 0.010).

Overall, dietary interventions including vegan and vegetarian diet, ketogenic diet, anti-inflammatory diet, elemental diet and fasting resulted in significant improvements in inflammatory markers. The inflammatory markers evaluated varied in the included studies and different diets seemed to have different effects on the inflammatory markers assessed. Although ESR and CRP were evaluated in the majority of the studies, only three studies reported significant improvements in these parameters.

#### 3.5.2. Clinical/Functional Measures

##### Pain

Pain is one of the major symptoms of RA [50] and was evaluated in ten studies. The change in pain following a dietary intervention was, however, not significant in eight out of the ten studies. In McKellar et al. [44], there was a significant improvement in pain following a Mediterranean diet as compared with habitual diet at 3 and 6 months (*p* = 0.011 and 0.04; respectively). Additionally, pain significantly decreased in participants who were in the fasting for 7–10 days followed by one year vegetarian diet group (*p* < 0.0001) [34].

##### Early Morning Stiffness (EMS)

EMS decreased significantly following a 4-week diet high in PUFAs and supplemented with fish oil (33 (7.34) to 22 (8.45) minutes; (*p* < 001) [43] and with fasting followed by one year vegetarian diet (*p* < 0.0002) [34]. McKellar et al. [44], reported significant improvement in EMS in the Mediterranean diet group as compared with habitual diet (*p* = 0.041).

##### Grip Strength

Grip strength significantly improved following a diet high in PUFAs and supplemented with fish oil providing 1.6 g EPA and 1.1 g DHA per day for 4 weeks (116 (13–26) to 136 (12–88) mmHg); (*p* < 001) [43]. Similarly, in Kavanagh et al.’s study, a 4-week elemental diet followed by reintroduction of food significantly improved grip strength (140.2 ± 96 to 155.9 ± 98.3 mmHg) (*p* = 0.008) [36]. Moreover, grip strength improved significantly with 7–10 days fasting followed by one year vegetarian diet (*p* < 0.0005) [34].

##### Ritchie’s Index

Significant reductions were found in five out of the six studies that evaluated Ritchie’s index. One study found that the decrease was seen after 6 months in the control group on habitual diet (12.5 to 10) (*p* < 0.05) [39]. Another study reported that Ritchie’s index decreased significantly in participants following an elemental diet followed by reintroduction of food for 4 weeks (12.6 ± 6.8 to 10.4 ± 7.2) (*p* = 0.006) [36]. Ritchie’s index also decreased significantly after 4 weeks in participants following a diet rich in PUFAs and supplemented with fish oil [33] and after 24 weeks of following a diet free from common allergenic foods (13.2 ± 4.4 to 9.2 ± 3.8) (*p* = 0.002) [40]. 7–10 days fasting followed by a vegetarian diet also significantly decreased Ritchie’s index (*p* < 0.0004) [34].

##### Disease Activity Score

In Sköldstam et al.’s study [42], participants following a Mediterranean diet showed a significant decrease in DAS28 (-0.56) (*p* < 0.001) and DAS28 was significantly lower in the Mediterranean diet group as compared to the habitual diet group (3.9 vs. 4.3) (*p* = 0.047). In another study, DAS28-ESR decreased significantly in participants following a diet rich in anti-inflammatory food (3.39 to 3.05) (*p* = 0.012) [35].

##### Number of Tender and Swollen Joints

The number of tender and swollen joints significantly decreased with (1) an elemental diet (*p* < 0.02 for tender joints and *p* = 0.04 for swollen joints) [28], (2) 7–10 days fasting followed by one-year vegetarian diet (*p* < 0.0002 for tender joints and *p* < 0.04 for swollen joints) [34], (3) an anti-inflammatory diet supplemented with 30 mg/kg fish oil (37% improvement *p* < 0.001 for tender joints) [25] and (4) a diet free from common allergenic foods (*p* = 0.04) [40]. Furthermore, the number of tender and swollen joints improved significantly in participants on an anti-inflammatory diet as compared to a Western diet (28% vs. 11%) and (34% vs. 22%) (*p* < 0.01), respectively [43].

##### Global Assessment

Global assessment differed significantly between the two study groups in two studies. In McKellar et al. [44], global assessment improved significantly in the Mediterranean diet group as compared with the habitual diet group at the end of the 6 months study period (*p* = 0.002). In a similar way, the global assessment improved significantly with 7–10 days fasting followed by one vegetarian diet (*p* < 0.0001) [34].

Data to assess the effects of dietary interventions on clinical and functional measures in RA is limited because dietary interventions and outcome measures evaluated varied among studies as did control diets. While several dietary interventions including anti-inflammatory diet, Mediterranean diet, elemental diet and vegetarian diet resulted in beneficial changes, it was not possible to determine whether a specific diet is associated with certain parameters. It did seem, however, that following a Mediterranean diet may result in significant improvements for more than one clinical/functional measure.

#### 3.5.3. Patient Questionnaires

Two questionnaires were used in four included studies. The health assessment questionnaire (HAQ) used to measure physical function [51] and the 36-item short form health survey designed to measure the health status (SF-36) [52]. HAQ improved significantly from 0.7 to 0.6 (*p* = 0.02) in participants following a Cretan Mediterranean diet [42] and 7–10 days fasting followed by a vegetarian diet for one year [34]. Additionally, two parameters in the SF-36; ‘vitality’ and ‘overall health compared to one year earlier improved following a Cretan Mediterranean diet + 11.3 (*p* = 0.018) and −0.6 (*p* = 0.016), respectively [42].

#### 3.5.4. Medications

Three studies reported on the dose of medications consumed by participants but only one study found a significant difference in the dose of corticosteroids taken pre- and post-intervention [43]. The dose of corticosteroids was significantly reduced for participants following both an anti-inflammatory diet (*p* = 0.022) and a Western diet when fish oil was supplemented (*p* = 0.027).

#### 3.5.5. Radiographs

Only one study reported on radiographic progression. Larsen score, number of erosions and the joint count in participants following both vegan and non-vegan diet significantly improved (*p* < 0.05) [41].

## 4. Discussion

The aim of this systematic review was to assess the effects of dietary interventions either alone or in combination with omega-3 supplementation for the management of RA. This review of twenty dietary intervention studies found considerable differences in terms of the types of dietary intervention and RA outcome measures evaluated. In summary, it was evident that the results were inconsistent across the included studies and did not provide a clear indication to support a specific dietary management strategy for RA. Results show that diets including Mediterranean, anti-inflammatory, vegan/vegetarian, elemental and allergenic-free diets have varying effects on RA outcome measures and do not always lead to improvements each time they are implemented in trials. Although the effects of the dietary interventions varied, this review demonstrated that dietary interventions in combination with omega-3 supplementation provided added benefits compared to diet alone in an RA population, albeit from one study. While dietary interventions seem to be promising to complement pharmacotherapy, the results of this review must be interpreted with caution and no conclusions must be drawn as to which diet is more effective for the management of RA. Although multiple outcome measures including HAQ, ESR, CRP, EMS, pain, grip strength and Ritchie’s index were evaluated in more than one trial, the implemented dietary interventions were distinct. Therefore, pooling of data for meta-analysis was inappropriate. To the author’s knowledge, this is the most up to date systematic review evaluating the effects of dietary interventions in RA and the first to explore explicitly the effects of dietary interventions either alone or in combination with omega-3 supplementation on a range of outcomes in RA.

Our findings are overall in keeping with conclusions from earlier reviews showing that the evidence on diet and RA is insufficient and inconclusive. The Cochrane review published in 2009 [15] included 15 studies and evaluated the effects of diets on three parameters in RA: pain, stiffness and physical function. The review by Nelson et al. [13] included studies on single food items, nutrients, dietary antioxidants and synbiotics and explored their effects on disease activity score only. Similarly, in Philippou et al.’s large systematic review [14], both diet and nutritional supplementation studies were considered and indications of their effects on RA outcomes were given. In contrast to other reviews, this review included an in-depth and detailed evaluation of the effects of whole diets, either alone or in combination with omega-3 PUFAs supplementation, on a range of outcome measures related to the management of RA.

People living with RA frequently experiment with different dietary approaches, seeking improvements in their symptoms. Rheumatologists often meet with people with RA who are trying out specific diets despite the lack of reliable data and evidence-based dietary guidelines for this population [53]. The included studies demonstrate that a vegetarian diet significantly improved various parameters of RA including pain, physical function, grip strength, early morning stiffness, number of tender and swollen joints and inflammatory status. A vegetarian-type dietary pattern, rich in fruits and vegetables, beans, nuts and seeds is thought to have favorable effects on inflammation due to the large amounts of fiber and anti-oxidants it contains [54]. Furthermore, significant improvements in disease activity score, physical function and overall health status were found with a Mediterranean diet which is similar to a vegetarian diet but with some animal protein included [42,44]. In the same way, vegan and anti-inflammatory diets improved several parameters of RA, predominantly those related to inflammation such as CRP, TNF α, T-cells, oxLDL, IgG and IgA [41,43]. The Mediterranean diet, anti-inflammatory diet, vegetarian and vegan diet have several key components in common and are similar in that they are all high in anti-oxidants, fibers, vitamins and minerals, all of which have anti-inflammatory properties and can alter the inflammatory processes and pathways in RA which is often reflected by improvements in symptoms.

The mechanisms for the influence of dietary interventions on the different parameters of RA, however, remain unclear [55]. Although several potential mechanisms through which diet may be related to pathways involved in RA exist, possible mechanisms were not highlighted in the majority of the included studies. Potential mechanisms through which diet can affect RA include amelioration of oxidative stress, alteration of the gut microbiota and reduction in inflammatory cytokines levels [56]. It is recognized that antioxidants found in fruit and vegetables aid in lowering oxidative stress levels and protect against the development of free radicals to prevent inflammation [57]. Additionally, research shows that oxidative stress may be associated with specific inflammatory biomarkers such as CRP and TNF-α in people with RA [58]. Findings from numerous studies suggest that a variety of dietary nutrients contain a range of anti-inflammatory properties, as such, diet is able modulate inflammatory biomarkers in the body [59]. The Mediterranean diet is recognized as an anti-inflammatory diet and has been linked to several health benefits [60]. Key components of the Mediterranean diet include extra virgin olive oil (EVOO), wholegrains, fish, fruits and vegetables. These elements can play significant anti-inflammatory roles by disrupting the arachidonic acid cascade, activity of immune cells and expression of pro-inflammatory genes [61,62,63]. In addition, the omega-3 PUFAs found in fish have been shown to influence the functions of lymphocyte and monocyte that are vital in the immune system’s ability to destroy invaders and are involved in the regulation of inflammatory pathways in chronic inflammatory diseases [17,64,65,66,67]. Furthermore, systematic reviews and meta-analyses on the Mediterranean diet have shown significant reductions in inflammatory biomarkers including CRP, ESR and IL-6 [68,69,70,71]. Given that inflammation is a fundamental mediator of RA pathogenesis, thus, the established association between a greater adherence to a Mediterranean dietary pattern and improved inflammatory status could explain some of the improvements seen in the trials that intervened with Mediterranean and an anti-inflammatory diets in this review.

Omega-3 PUFAs are well-known to have anti-inflammatory properties and can reduce inflammation by elevating autophagy in macrophages [72]. The use of PUFAs particularly omega-3 PUFAs for the management of RA has been researched since the mid 80′s. Numerous clinical trials have demonstrated improvements in the RA symptoms of pain and early morning stiffness as well as reductions in the dose of NSAIDs consumed with increased intakes of omega-3 PUFAs [21]. Although omega-3 PUFAs are the most comprehensively studied dietary supplements in RA, most studies of omega-3 PUFAs supplementation have not considered combining the supplementation with a dietary intervention. Only two studies evaluated the effectiveness of a dietary intervention in combination with omega-3 supplementation. Adam et al. [43] reported significant reductions in CRP, TNF- α and dose of corticosteroids in participants following both an anti-inflammatory diet and a Western diet with fish oil supplements. Furthermore, significant improvements in the number of tender joints and LTB4 levels were found in the anti-inflammatory diet group only when omega-3 PUFAs was supplemented. In the second study, a diet rich in PUFAs (P:S ratio 5:0) in combination with omega-3 supplementation (1.6 g EPA/d and 1.1 g DHA/d) resulted in significant benefits in terms of EMS, Ritchie’s index and grip strength (33). This indicates that the anti-inflammatory benefits of a diet in combination with omega 3 PUFAs may be superior to the diet alone and further research in this area is warranted.

People living with RA also often report intolerances to certain foods and claim that their symptoms are aggravated by specific foods such as red meat, gluten-containing foods, foods that are high in sugar and alcohol [73]. It has been suggested that the elimination of certain food items from the diet may help alleviate RA symptoms. Yet, the evidence on the effects of elimination diets on RA is limited. Panush et al. [74] suggested that hypersensitivity responses may be provoked by foods from protein sources leading to aggravation of RA symptoms. Though there is evidence on the pro-inflammatory effects of diets high in processed and red meats [75], Panush et al. [31] found no significant differences in RA symptoms when red meat was removed from the diet. Gluten has also been to associated with increased inflammation in the body and studies have shown that people living with RA may be at higher risk of celiac disease [76,77]. In one trial [40], significant improvements in Ritchie’s index and the number of tender and swollen joints were found when wheat was eliminated from the diet. Another three studies included in this review [34,41,45] intervened with gluten free vegan diet reported significant improvements in several RA parameters; however, it remains unclear as to whether improvements were observed as a result of gluten elimination or the vegan diet itself.

This systematic review is limited by the heterogeneity among the included studies, particularly when it comes to the outcome measures. Furthermore, the lack of standardization of outcome measures, types of dietary interventions, comparators and methodologies made it difficult to synthesize studies and thus, draw conclusions. In addition to the discrepancies found between studies, which rendered a meta-analysis unfeasible, the presence of several levels and qualities of evidence is also to be considered as a potential limitation of the evidence presented in this systematic review. The majority of the studies were deemed to be of unclear or high risk of bias and did not control for confounding factors such as the severity of RA and presence of comorbidities. It is also worth noting that due to the nature of dietary interventions, double-blind studies are not always achievable, hence, increasing the risk of bias in the included studies. Additionally, the literature search for this review excluded grey literature and studies written in non-English language. The strength of this systematic review is the robust methodology that included explicit eligibility criteria and an extensive and comprehensive database search. A broad search strategy was used during the literature search to guarantee all aspects related to the review’s aims and objectives were included. In addition, this review is the first of its kind to consider the combined effects of diet and omega-3 supplementation.

The disparity of results across the studies exploring the effectiveness of dietary interventions highlights that further research is needed in this area in order to draw definitive conclusions as to which diet is best for disease management in adults with RA. Both short- and long-term effects of various dietary interventions on important RA measures need to be specifically investigated. Studies with adequate long-term follow-up and larger sample sizes are required for future research.

Pre-defined and standardized outcome measures should be used to determine whether dietary interventions are effective for the management of RA. The Outcome Measures in Rheumatology (OMERACT) is an internationally organized group of experts that aims to develop optimal core outcome measures for use and reporting in clinical trials involving people with rheumatic and musculoskeletal conditions [78]. Future research should consider using the core domains agreed upon by the OMERACT group along with other relevant measures. This will help in interpreting results and will allow comparisons of findings across various studies. Moreover, future research on the effect of dietary interventions should consider including quality of life as an outcome measure since there is growing evidence showing that lifestyle changes including dietary changes may have an influence and are positively associated with quality of life in adults living with RA [79].

The mode of delivery of the dietary interventions is also worthy of consideration. Digital health interventions are increasingly being used to support independent self-care for people living with chronic conditions and has been amplified since the COVID-19 pandemic. There is a growing body of evidence on the advantages of using digital health interventions and its benefits for the management of chronic diseases [80,81]. According to Dietitians Australia position statement, nutrition consultations delivered through telehealth methods were found to be as effective as face-to-face consultation [82]. As such, telehealth methods may be considered in future dietary intervention studies in this population, either as a complement or as a replacement to face-to-face services. Dietetic and multidisciplinary supports are severely lacking for services in the management of RA.

In addition, none of the included studies used a theory-based intervention. Theory-based interventions provide a better understanding of the processes and effectiveness of interventions through the identification of key concepts and hypotheses that are related to behavioural change techniques that can be used for designing the intervention [83]. Priorities for future trials should also include strategies to enhance transparency. This involves accurately reporting intervention content, delivery method and procedures for maintaining behavioural change. Often with dietary intervention trials, a change in an individual’s dietary habits and lifestyle is required [84]; therefore, it is essential to monitor compliance and to ensure individuals remain highly motivated.

## 5. Conclusions

The role and efficacy of dietary interventions in the management of RA remains uncertain. Certain dietary interventions including anti-inflammatory diets, vegetarian diets, Mediterranean diet and elemental diets may help alleviate the symptoms significantly, others mildly, whereas it seems that a certain group of foods continue to aggravate the symptoms. Despite the numerous studies that have been conducted in this area, there remains much heterogeneity and bias across both interventions and results of the clinical trials. The way forward remains the performance of trials under rigorously controlled conditions with reasoned extrapolation of data when it comes to interpreting the results.

## Figures and Tables

**Figure 1 nutrients-13-03506-f001:**
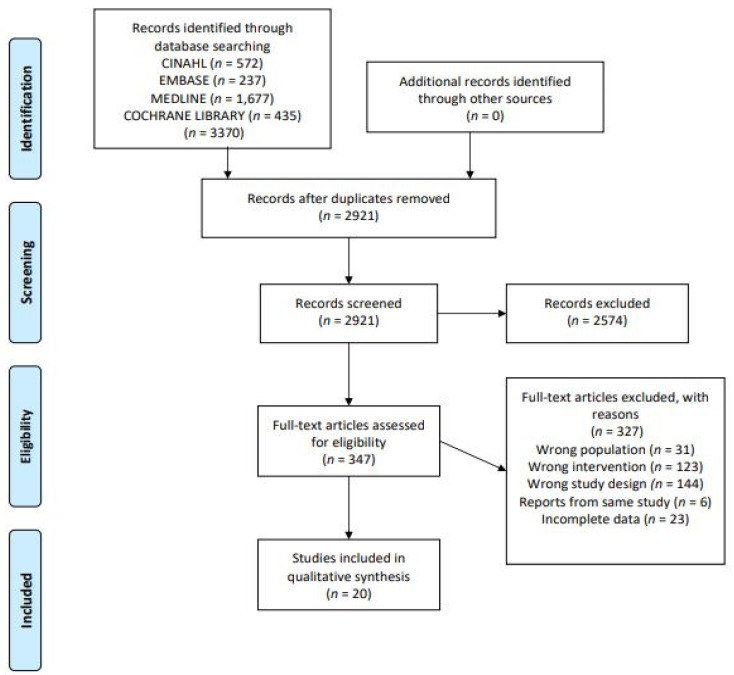
PRISMA flow diagram of study selection.

**Figure 2 nutrients-13-03506-f002:**
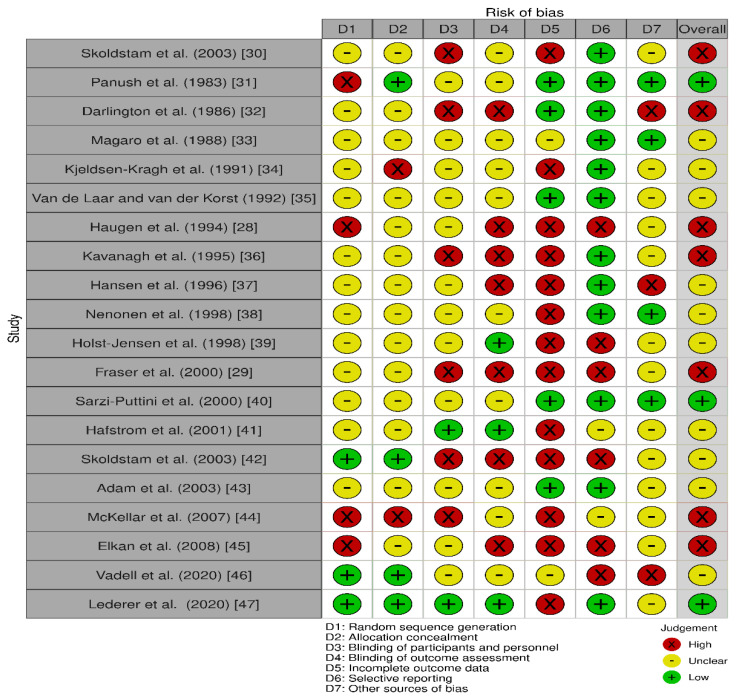
Risk of bias summary: review authors’ judgments about each risk of bias item for each included study.

**Table 1 nutrients-13-03506-t001:** PICOS criteria for inclusion and exclusion of studies.

PICOS	Inclusion/Exclusion Criteria
**Population**	Inclusion: Adults with a definite diagnosis of RA according to a specified diagnostic criterion Exclusion: All animal and pediatric studies, or studies involving women who are pregnant/breastfeeding
**Intervention**	Inclusion: - Studies that compared a dietary intervention to an alternative diet or control (habitual diet) - Studies that included omega-3 supplementation alongside a dietary intervention (eg, omega-3 supplementation plus dietary intervention vs. comparator) - Studies that compared a dietary intervention to fasting - Studies that intervened with fasting followed by another diet Exclusion: - Studies that included a dietary intervention alongside a co-intervention such as physical activity, behavioral training, or other lifestyle interventions - Studies that intervened with only supplements or drugs - Studies that implemented fasting only or compared fasting to habitual diet
**Comparator**	Inclusion: - Control group (habitual diet) - Another dietary intervention Exclusion: - Studies without a comparator group - Studies that compared an intervention with a drug or a supplement
**Outcomes**	Inclusion: Studies that reported the effect of diet with or without omega-3 supplementation on aspects of RA management, i.e., symptom control, clinical and biochemical measures, disease activity Exclusion: Studies that investigated the effect of a dietary intervention on the risk of developing RA were not included.
**Study design**	Inclusion: The current systematic review included randomized and non-randomized controlled trials. Publications were eligible if they were published in peer-reviewed scientific journals and were written in English language. Exclusion: Reviews, cohort studies, cross-sectional studies, case–control studies, conference abstracts, editorials, letters, and reviews.

**Table 2 nutrients-13-03506-t002:** Summary of study characteristics for included studies.

Author (Year)	Country	Study Design	Participants’ Characteristics	Intervention	Comparator	Duration	Primary Outcome Measures	Results (Post- Intervention Changes)	
								Within Group	Between Groups
Skoldstam et al. (1979) [30]	Sweden	RCT	*n* = 26 Mean age: 53 yrs Sex: 73% f	7–10 days fasting followed by 9-week lactovegetarian diet	Habitual diet	10 weeks	Pain, EMS, dose of NSAIDs	NSD	
Panush et al. (1983) [31]	US	RCT	*n* = 33 Mean age: 55 yrs Sex: 34.6% f	Diet free of additives, preservatives, fruit, red meat, herbs, and dairy products	Placebo diet	10 weeks	EMS, number of tender and swollen joints, grip strength, patient and examiner assessment, walk time, ESR, RF, Hct/C3/C4	NSD	
Darlington et al. (1986) [32]	UK	RCT	*n* = 45 Mean age: not reported Sex: 89% f	Elimination diet Week 1: tolerated foods followed by reintroduction of foods that are unlikely to cause intolerance followed by habitual diet	Habitual diet	6 weeks	Pain, EMS, grip strength, number of painful joints	NSD	Inadequate reporting
Magaro et al. (1988) [33]	Italy	RCT	*n* = 12 Mean age: Group A: 37 yrs Group B: 36 yrs Sex: 100% f	Group B: Diet high in PUFAs (P:S ratio 5:0) + fish oil supplement (1.6 g EPA/d and 1.1 g DHA/d)	Group A: Diet high in saturated fatty acids (P:S ratio 1:33)	4 weeks	DAS28, neutrophil chemiluminescence, Ritchie’s index, EMS, grip strength	Significant improvements in Group B: Ritchie’s inde X(17.2 (3.38) to 10.6 (3.48)); (*p* < 001), EMS (33 (7.34) to 22 (8.45)) mins; (*p* < 001); Grip strength (116 (13–26) to 136 (12–88)) mmHg; (*p* < 001)	Significant differences in: Ritchie’s index (Group B: 10.6 (3.48) vs. Group A: 21–4 (3.2); (*p* < 0.005) EMS (Group B: 22 (8.45) vs. Group A:36 (10.17) minutes; (*p* < 0.01) Grip strength (Group B:136 (12–88) vs. Group A: 104 (21–58) mmHg; (*p* < 0.01)
Kjeldsen-Kragh et al. (1991) [34]	Norway	RCT	*n* = 53 Mean age: 4.5 years Sex: 85% f	7–10 days: fasting followed by 3·5 months: gluten-free vegan diet followed by 9 months: vegetarian diet	Habitual diet	13 months	Grip strength, Ritchie index, EMS, Global assessment, Number of tender and swollen joints, pain HAQ, ESR, CRP, white blood cells/platelet count	Significant improvements in the intervention group for: Grip strength (*p* < 0.0005), Ritchie Index (*p* < 0.0004), EMS (*p* < 0.0002); Number of tender joints (*p* < 0.0002), Number of swollen joints (*p* < 0.04), Pain (VAS) (*p* < 0.0001 for intervention group and *p* < 0.02 for control), HAQ (*p* < 0.0001), ESR (*p* < 0.002), CRP (*p* < 0.005) White blood cells/platelet count decreased significantly in the intervention group (*p* < 0.0010) and in the control group (*p* < 0.006)	Significant improvement in the intervention group as compared with control for: Grip strength (*p* < 0.02), Ritchie index (*p* < 0.0004), EMS (*p* < 0.0001), Global assessment (*p* < 0.0001), Number of tender joints (*p* < 0.0001), Number of swollen joints (*p* < 0.02), pain (*p* < 0.02), HAQ (*p* < 0.0001), ESR (*p* < 0.001), CRP (*p* < 0.0001)
Van de Laar and van der Korst (1992) [35]	Netherlands	RCT	*n* = 94 Mean age: 58 yrs Sex: 70% f	Allergen free diet	Allergen restricted diet	12 weeks	EMS, number of tender and swollen joints, Ritchie’s index, grip strength, global assessment, ESR, CRP, walking time	Significant decrease in body weight in the allergen free diet group (*p* = 0.016)	NSD
Haugen et al. (1994) [28]	Norway	NRCT	*n* = 17 Mean age: 50 yrs Sex: 80% f	Elemental diet (E028)	Soup that included: milk, meat, fish, shellfish, orange, pineapples, tomatoes, peas and flour of wheat and corn	3 weeks	Ritchie‘s index, number of tender and swollen joints, grip strength, EMS, pain, ESR, CRP, hemoglobin, albumin and erythrocyte count, global assessment	Number of tender joints decreased significantly in the intervention group (*p* = 0.04) ESR and thrombocyte count improved in the control group (*p* = 0.03) and (*p* = 0.02), respectively	NSD
Kavanagh et al. (1995) [36]	UK	RCT	*n* = 47 Mean age: 45.6 yrs Sex: 78.7% f	E028 followed by reintroduction of food	Habitual diet with E028	4 weeks	ESR, CRP, Ritchie’s index, thermographic score, grip strength, functional score	Significant improvements in the intervention group for: Ritchie’s index (12.6 ± 6.8 to 10.4 ± 7.2) (*p* = 0.006), Grip strength (140.2 ± 96 to 155.9 ± 98.3 mmHg) (*p* = 0.008)	NSD
Hansen et al. (1996) [37]	Denmark	RCT	*n* = 109 Mean age:57 yrs Sex: 74.6% f	Graastener diet: 20–30% fat, 1.5 g/kg BW protein, 800 g fresh fish per week	Habitual diet	4 months	Number of tender and swollen joints, pain, HAQ, Global assessment, acute phase reactant, X-ray, EMS	Authors state: ‘Significant improvement in the duration of morning stiffness, number of swollen joints, pain status’	NSD
Nenonen et al. (1998) [38]	Finland	RCT	*n* = 43 Mean age:53 yrs Sex: 83% f	Uncooked vegan diet	Habitual diet	3 months	Pain, number of swollen joints, number of tender joints, EMS, HAQ, Ritchie’s index, CRP, ESR	NSD	
Holst-Jensen et al. (1998) [39]	Denmark	RCT	*n* = 30 Mean age: 49.5 yrs Sex: 80% f	Commerical liquid elemental diet (top upTM Standard, Ferrosan Ltd., Denmark)	Habitual diet	4 months	EMS, HAQ, number of swollen joints, pain, Ritchie’s index, global assessment, ESR	EMS decreased significantly in the control group (3.5 to 2.5 min) (*p* < 0.05) Ritchie’s inde Xdecreased significantly in the control group (12.5 to 10) (*p* < 0.05)	Significant reductions in the intervention group as compared with control for: Number of tender joints (7 vs. 9) (*p* = 0.006), ESR (40 vs. 47 mm/h) (*p* = 0.018)
Fraser et al. (2000) [29]	Norway	NRCT	*n* = 23 Fasting group: Mean age: 49 yrs, Sex: 90% f Ketogenic group: Mean age:44 yrs, Sex: 92% f	7-day ketogenic diet	7-day fast	1 week	IL-6, DHEAS	IL-6 decreased significantly after fasting for 7 days (35.5 to 22.5 pg/mL) (*p* < 0.05) DHEAS increased significantly after fasting for 7 days (3.28 to 4.40 mmol/L) (*p* < 0.01) and after a 7-day ketogenic diet group (2.42 to 3.23 mmol/L) (*p* < 0.01)	Not reported
Sarzi-Puttini et al. (2000) [40]	Italy	RCT	*n* = 50 Mean age:50 yrs Sex: 78% f	Diet free from: wheat meal, eggs, milk, strawberries and acid fruit, tomato, chocolate, crustacean, dried fruit Lean cuts of red meat allowed	Diet containing common allergenic foods	24 weeks	EMS, HAQ, number of tender and swollen joints, pain, Ritchie’s index	Number of tender and swollen joints decreased significantly in the intervention group (9.5 ± 4.1 to 7.1 ± 3.2) (*p* = 0.031) and (6.4 ± 3.1 to 5.1 ± 2.3) (*p* = 0.002), respectively Ritchie’s inde Xdecreased significantly in the intervention group (13.2 ± 4.4 to 9.2 ± 3.8) (*p* = 0.002)	Not reported
Hafstrom et al. (2001) [41]	Sweden	RCT	*n* = 66 Mean age: 50 yrs Sex: not reported	Gluten free vegan diet	Well-balanced non-vegan diet	12 months	IgG, IgA, radiographic progression	IgG anti-gliadin decreased significantly in the vegan diet group (5 to 2) (*p* = 0.0183) IgA anti-gliadin decreased significantly in the non-vegan diet group (14.5 to 12.5) (*p* = 0.0201) Modified Larsen score, number of erosions and the joint count improved significantly in both groups	NSD
Skoldstam et al. (2003) [42]	Sweden	RCT	*n* = 56 Mean age: 58.5 yrs Sex: = 82% f	Cretan Mediterranean diet (MD)	Habitual diet (HD)	12 weeks	DAS 28, HAQ, SF-36, dose of NSAIDs	DAS28 decreased significantly in MD group (4.4 to 3.9) (*p* < 0.001) HAQ decreased significantly in MD group (0.7 to 0.6) (*p* = 0.02) Improvement in vitality (+11.3) (*p* = 0.018) and overall health compared to one year earlier (−0.6) (*p* = 0.016) in the SF- 36 in MD group	Significant improvements in MD group as compared to control group for: DAS28 (3.9 for MD vs. 4.3 for control) (*p* = 0.047) HAQ: (0.6 for MD vs. 0.8 for control) (*p* = 0.012)
Adam et al. (2003) [43]	Germany	RCT Double-blind crossover	*n* = 68 Mean age: 57.4 ± 12.8 yrs Sex: 93.3% f	Anti-inflammatory diet (AID) Patients in both diet groups were assigned to receive either placebo or fish oil capsules (30 mg/kg body weight)	Western diet (WD)	6 months	Global assessment, pain, grip strength, EMS, HAQ, Number of tender and swollen joints, blood cells, cytokines, eicosanoids, dose of Corticosteroids, CRP, LBT4, TNF-α	CRP decreased significantly for individuals in both WD and AID groups who are on methotrexate when fish oil was supplemented (2.03 ± 1.8 mg/dL vs. 1.69 ± 1.5 mg/dL) (*p* < 0.05) Number of tender joints improved significantly in AID group when fish oil was supplemented in months 5,6,7,8 (37% improvement) (*p* < 0.001) LTB4 decreased significantly in AID group when fish oil was supplemented for 3 months (*p* = 0.009) Dose of corticosteroid decreased significantly in both WD and AID groups after 3 months of fish oil supplementation (*p* = 0.027 for WD group, *p* = 0.022 for AID group) TNF-α decreased significantly in both WD and AID groups when fish oil was supplemented for months 6,7, 8 (*p* = 0.004)	The number of tender and swollen improved significantly in the AID group as compared to WD group (28% vs. 11%) and (34% vs. 22%) (*p* < 0.01), respectively Patients’ and physicians’ global assessment of disease activity and patients’ assessments of pain improved significantly more in the AID group as compared to WD group (*p* < 0.05)
McKellar et al. (2007) [44]	Scotland	RCT	*n* = 130 Mean age: 54 yrs Sex: 100% f	Mediterranean diet (MD)	Healthy diet	5 months	Number of tender and swollen joints, patient global assessment, pain, EMS, DAS28, HAQ, ESR, CRP, IL-6	Not reported	Significant improvements in the intervention group as compared with the control group for: patient global assessment (*p* = 0.002), pain (*p* = 0.049) and EMS (*p* = 0.041)
Elkan et al. (2008) [45]	Sweden	RCT	*n* = 58 Vegan group: Mean age: 49.9 yrs, 93.3% f Non-vegan group Mean age:50.8 yrs, 85.6% f	Gluten- free vegan diet	Well-balanced non-vegan diet	12 months	oxLDL, anti-PCs	OxLDL decreased Significantly in the vegan diet group (54.7 to 48.6) (*p* = 0.09)	Anti-PC IgM was significantly higher in vegan group (F = 8.0, *p* = 0.0006)
Vadell et al. (2020) [46]	Sweden	RCT	*n* = 50 Mean age: 61 ± 12 yrs Sex: 77% f	Diet rich in anti-inflammatory foods	Habitual diet	10 weeks	DAS28-ESR	DAS28-ESR decreased significantly in the intervention group (3.39 to 3.05) (*p* = 0.012)	NSD
Lederer et al. (2020) [47]	Germany	RCT	*n* = 53 Mean age: 31 yrs Sex: 63% f	Vegan diet (VD)	Meat rich diet	5 weeks	Sialylated antibodies, percentage of regulatory T-cells, IL-10	Significant improvement in: Sialylated antibodies in VD (0.8 ± 0.4 to 1.4 ± 1.4) (*p* = 0.023) and in the meat rich group (0.9 ± 0.5 to 1.6 ± 1.2) (*p* = 0.010) T-cells in VD group (6.0 ± 1.7% to 7.1 ± 1.9%) (*p* < 0.001) and in meat rich group (6.3 ± 2.2% to 7.7 ± 2.4%) (*p* < 0.001)	NSD

**Abbreviations**: VAS: Visual Analogue Scale; EMS: Duration of early morning stiffness; NSAIDs: Non-steroidal anti-inflammatory drugs; RF: Rhheumatoid factor; Hct: Hematocrit; C3: Complement component 3; C4: Complement component 4; PUFA: Polyunsaturated fatty acids; EPA: Eicosapentaenoic acid; DHA: Docosahexaenoic acid; DAS28: Disease activity score in 28 joints; HAQ: Health assessment questionnaire; CRP: C-reactive protein; ESR: Erythrocyte sedimentation rate; IL-6:Interleukin-6; IL-10: Interleukin-10; DHEAS: Dehydroepiandrosterone sulfate; SF-36: Short form health survey; TNF- α: Tumor necrosis factor alpha; oxLDL: Oxidized low-density lipoprotein; anti-PCs: Immunoglobulin M antibodies against phosphorylcholine; BMI: Body mass index; BW: Body weight; f: females; yrs: years; mins: minutes; NSD: No significant difference.

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
