# Peer review of "Dietary Interventions with or without Omega-3 Supplementation for the Management of Rheumatoid Arthritis: A Systematic Review"

_nutrients, 2021, doi:10.3390/nu13103506_

Round 1

Reviewer 1 Report

Authors presnet their systematic review about dietary interventions with or without omega-3 supplementation for the management of rheumatoid arthritis. Review  is well writen but on the begining Table 2. Summary of study characteristics for included studies have to be changed. It is too big and it distorts the view of the whole work, distracting attention from other important criteria presented in the studie. Due the fact that the role and efficacy of dietary interventions in the management of RA remains uncertain souch systematic review seems to be interesting and should be done. But some minor changes should be included in the test to maintain the scientific quality of the publication. Specialu way of Inflammatory markers and Clinical/ functional measures presentation. Significant simplification of the determined parameters does not allow to draw precise conclusions, and thus the main goal of the work is blurred. It should be rewriten to maintain appropriate Jurnal criteria. 

Reviewer 2 Report

The article by Tala Raad and colleagues is a well-written and rigorously conducted systematic review addressing the effectiveness of omega-3-PUFA-enriched diets in improving different clinical outcomes and inflammatory markers in patients with rheumatoid arthritis. I don’t have major criticisms. Only minor edits are needed before considering the manuscript suitable for publication.

  • Line 46 -“co-morbid conditions”: amend into “comorbid conditions”
  • -Line 76: “has been researched for many years” I suggest to quote here this paper : PMID: 32204518
  • Table 1: amend “Exclusion: All animal and paediatric studies, or studies...”
  • Table 1: write in bold character the terms in the first column, and underline “Exclusion” and “Inclusion” in each row of the second column
  • Lines 130-134: insert commas between all different points: e.g country, (3)…
  • Resolution of figur 1 should be improved
  • Table 2: use bold text for the terms in the first row
  • Table 2: azo colour-ings : food colourings?
  • “number of erosions and the joint count improved significantly in both groups”
  • Lines 177-179: the countries mentioned are nine, not ten
  • -Line 203: amend “ providing 1.6 g EPA and 1.1 g DHA per day”
  • -Line 244: please clarify reference after Adam et al., which is McKellar
  • -Line 262: “increased”
  • Line 269: “a Mediterranean diet”
  • Lines 281-283: please improve the structure of this sentence.
  • Line 295: “Sköldstam “
  • Line 353: “explicitly the effects”
  • Line 381: fibers
  • Line 392: shows
  • Line 398: can play
  • Line 403: I suggest to quote here these papers: PMID: 32204518 + PMID: 31816979 + PMID: 34255301 + PMID: 31547601
  • Line 427: omega-3 PUFAs
  • Line 438: amend “been to associated to” into “been associated with”
  • Line 443:amend: “parameters; however, it remains unclear whether improvements were observed as a result of gluten elimination or the vegan diet itself.”
  • Line 454: interventions, double-blind…
  • Line 458: included explicit eligibility criteria
  • Table 2 should be better formatted; please avoid to split single terms into two different lines
